# Use of Unmanned Aerial Vehicle Imagery and Deep Learning UNet to Extract Rice Lodging

**DOI:** 10.3390/s19183859

**Published:** 2019-09-06

**Authors:** Xin Zhao, Yitong Yuan, Mengdie Song, Yang Ding, Fenfang Lin, Dong Liang, Dongyan Zhang

**Affiliations:** 1National Engineering Research Center for Agro-Ecological Big Data Analysis & Application, Anhui University, Hefei 230601, China; 2Faculty of International Trade, Shanxi University of Finance and Economics, Taiyuan 030006, China; 3School of Geography and Remote Sensing, Nanjing University of Information Science & Technology, Nanjing 210044, China

**Keywords:** rice lodging, UAV, UNet, semantic segmentation, assessment, *Oryza sativa* L.

## Abstract

Rice lodging severely affects harvest yield. Traditional evaluation methods and manual on-site measurement are found to be time-consuming, labor-intensive, and cost-intensive. In this study, a new method for rice lodging assessment based on a deep learning UNet (U-shaped Network) architecture was proposed. The UAV (unmanned aerial vehicle) equipped with a high-resolution digital camera and a three-band multispectral camera synchronously was used to collect lodged and non-lodged rice images at an altitude of 100 m. After splicing and cropping the original images, the datasets with the lodged and non-lodged rice image samples were established by augmenting for building a UNet model. The research results showed that the dice coefficients in RGB (Red, Green and Blue) image and multispectral image test set were 0.9442 and 0.9284, respectively. The rice lodging recognition effect using the RGB images without feature extraction is better than that of multispectral images. The findings of this study are useful for rice lodging investigations by different optical sensors, which can provide an important method for large-area, high-efficiency, and low-cost rice lodging monitoring research.

## 1. Introduction

The *Oryza sativa* L. rice is one of the world’s three major food crops besides from wheat and maize, and its stable production has a major impact on world politics and economics [1]. Lodging weakens rice photosynthesis, decreases yield [2,3], and causes difficulties in harvesting [4,5], which further reduces rice quality [6]. During rainy weather, water increases in the area, which easily causes mildew in rice and severely affects rice quality and food safety. In rice planting management and follow-up agricultural insurance policy development, the monitoring and analysis of the lodging situation is particularly important. Traditional manual measurement methods require agricultural personnel to conduct measurement and sampling analysis in the field. Such methods are time-consuming, labor-intensive, and inefficient [7]. They may cause secondary damage to crop during measurement. Therefore, an efficient method for obtaining crop lodging information should be developed.

Satellite remote sensing technology has a increasing application in the extraction of crop lodging information [8,9,10]. However, with limitations of its temporal and spatial resolution, the technology is not a real-time and accurate method of monitoring ricelodging information. In addition, satellite imageacquisition is vulnerable to cloud.

An unmanned aerial vehicle (UAV) is controllable and capable of carrying multiple sensors for performing multiple missions. Moreover, it has low cost, high time efficiency, high resolution, and fly in cloudy conditions. UAVs are suitable for crop lodging monitoring and analysis. They can be equipped with various sensors, such as high-resolution digital, multispectral, hyperspectral, and infrared cameras, LiDAR (Light Detection and Ranging), and other similar devices, which can provide many physiological information of lodged crops. By comparing the color and texture characteristics of lodged and non-lodged corn images, Li Zongnan [11] selected characteristics favorable to extract the area of lodged corn. The error of extracting lodged corn area based on the red, green, and blue mean texture features is 0.3%, and the maximum is 6.9%. Yang et al. [12] proposed a spatial and spectral mixed image classification method to determine the lodging rate of rice, which has the accuracy of 96.17%.

With an increase in computer computing power and the development of deep learning network architecture, deep learning technology in agricultural disease assessment [13,14], variety classification [15,16,17], weed identification [18], and crop count [19,20] have achieved evident results. Ferentinos et al. [21] built convolutional neural network models to perform plant disease detection and diagnosis, and obtained the high accuracy of 99.53%. Uzal et al. [22] used a convolutional neural network for estimating the number of seeds into soybean pods and obtained 86.2% accuracy in the test. But deep learning techniques have few applicaiton in evaluating rice lodging.

In this study, a UAV equipped with high-definition digital and multispectral cameras was used to obtain rice lodging canopy images. After data were pre-processed and augmented to generate datasets, the UNet model was trained to study the effect of lodging information extraction by common digital and multispectral images. This study also discussed the influence of different characteristic parameters and sensors on the recognition accuracy of rice lodging. These can provide technical support for the extraction and evaluation of crop lodging based on a drone platform.

## 2. Materials and Methods

### 2.1. Study Area

The experimental site of the study is located at Qixing Farm, Sanjiang Management Bureau, Heilongjiang Province. Qixing Farm is located on the hinterland of Sanjiang Plain, as shown in Figure 1. The terrain is low and flat next to four large ports in Fujin, Tongjiang, Fuyuan, and Rao River. The terrain has a cultivated area of 813 hectares, including 700 hectares of rice, which is the largest farm for rice cultivation and production in the reclamation area. The study rice variety is *Longjing 32*, and the growth period is at the ripening phase.

### 2.2. Data Acquisition

In this study, the DJI Phantom 4 UAV platform equipped with high resolution digital and multispectral cameras was used. The digital camera has a resolution of 4000 × 3000 pixels and a field of view (FOV) of 94°. The multispectral camera is a Survey 3 camera produced by MAPIR CAMERA, which includes red (660 nm), green (550 nm), and near-infrared (850 nm) images, referred to as Red, Green, NIR (RGN). As it weighs 50 g, it’s easy to integrate into the UAV. It took a 2 s interval, the ISO was 100, exposure time was 1/1000 s and it saved in 12-bit RAW format. It has an image resolution of 4000 × 3000 pixels and a field of view (FOV) of 87°. Images were taken on September 28, 2018, at 2:30 pm, when the weather was fine with few clouds, and the wind speed is around 2 m/s.

An automatic route planning flight was performed using DJI GS Pro, covering an area of 7.97 hectares. The resolution of image taken by the UAV at a height of 50 m is 2.2 cm/pixel, but it needs more than 30 min and 3 flights to complete all the missions. Environmental factors such as illumination, wind speed and wind direction easily changed from 0.5 to 1 h and affected the image quality. When the flying height was at 150 m, it had a low spatial resolution. Considering the duration of the battery, spatial resolution, natural environment and other factors, the height of 100 m is selected as the optimal flying height.

### 2.3. Data Preprocessing

As shown in Figure 2, the images including ordinary RGB and multispectral images require data preprocessing. The RGB images could be directly spliced with Agisoft Photoscan developed by Agisoft LLC and then used PhotoShop CC developed by Adobe to complete image cropping. Multispectral images were in a 12-bit RAW format and captured in two-second interval. However, most of these images were useless. Therefore, a Survey 3 camera software was used to convert the format and filter the converted images.

### 2.4. Dataset Generation

After data were pre-processed, a single RGB image and a single multispectral image of the entire study area was obtained. With the subsequent model training requirement, the images were augmented to generate datasets. The single image was randomly cut into small images at a resolution of 320 × 320 pixels. These small images were then flipped, rotated, scaled, sheared, and skewed randomly to enhance the general performance of the dataset. Data augmentation generated 5000 small images in a sequence for the two original large images, and the obtained data were randomly divided into training dataset, validation set, and test set at a ratio of 0.7:0.15:0.15. The manually processed ground image (also named as ground truth image) dataset was manually segmented by an expert using the LabelMe software. That is, each image has a binary hand-drawn mask indicating the rice lodging area in white. Some sample images are shown in Figure 3.

### 2.5. Model Establishment and Evaluation

The UNet is a deep learning network architecture, which is widely used for semantic segmentation tasks. It was first proposed by Olaf Ronneberger [23] for medical image segmentation. Unlike a general convolutional network, UNet is a full-convolution network that does not include a fully connected layer and is not demanding on the amount of dataset. This network is simple, efficient, and easily used. Its architecture was mainly divided into two parts, namely, an encoder and a decoder. The encoder continuously sampled through multiple convolution layers to obtain different image feature levels. The decoder performed multi-layer deconvolution on the top-level feature map and combined different feature levels in the down-sampling process to restore the feature map to the original input image size and complete the end-to-end semantic segmentation task of the image.

In this study, the network was improved to make it suitable for the segmentation of rice lodging canopy images based on the original UNet architecture. The architecture is shown in Figure 4. Conv is a convolutional layer (convolution kernel is 3 × 3 and padding is 1), BN is batch normalization, Relu is the activation layer, Maxpool is the maximum pooling layer, and UpConv is the transposed convolution. The input image is a three-channel image with a resolution of 320 × 320 pixels, and the output is a single-channel segmented image. The left and right halves of the network architecture are the encoder and the decoder, respectively. The encoder structure is consistent with the common convolutional neural network. The input image initially generates a 64-channel feature map through the double Conv+BN+Relu group and subsequently generates a 1024-channel feature map, whose ratio is 1/16 of the original image (i.e., 20 × 20 pixels) through a four-time continuous Maxpool+(Conv+BN+Relu) × 2 group. The decoder contains four consecutive UpConv+(Conv+BN+Relu) × 2 combination operations and samples the 1024-dimensional feature map at 20 × 20 pixels to the 64-channel feature map at the original size (i.e., 320 × 320 pixels). The intermediate feature map generated by the encoder is concatenated to the feature map generated by the decoder by using Skip Connect on the way. Lastly, a segmentation image is generated by a convolution layer.

The model evaluation adopts the dice coefficient, which is a commonly used evaluation metric in the image segmentation field. The calculation formula is shown in Equation (1). The meaning is the ratio of the intersection of the two images to the total area, ranging from 0 to 1. The closer the dice coefficient is to 1, the better the model effect will be.
(1)$$dice=\frac{2\left(R_{seg}\cap R_{gt}\right)}{R_{seg}+R_{gt}}$$
$R_{seg}$ is the result of the segmentation and $R_{gt}$ is the result of the manual labelling.

The UNet model training algorithm was implemented with Python 3.6 in Spyder software, and code can be found at the URL “https://github.com/zhxsking/unet_on_jsj”. The PyTorch library was used to build the UNet. The code runs on a 64-bit system in Windows 10, comprising a CPU with Intel Xeon E5-2630 v4, a 128 G memory, and an NVIDIA Geforce RTX 2080Ti(11 G) GPU. In the actual experiment, an Adam optimization algorithm was used. The loss function involved the use of the cross entropy, which was commonly utilized in the segmentation field. The learning rate was set to $1\times 10^{-4}$, which could improve the results. In addition, the strategy of halving the learning rate for every 10 epochs was adopted in the training process. Thus, the initial learning rate was set to $5\times 10^{-4}$, and the mini-batch size was set to 12 under the condition of maximizing the use of 11 G memory. A total of 100 epochs were cycled.

Proper weight initialization is critical for deep neural networks. Glorot et al. [24] showed that inappropriate weight initialization causes network gradient explosion or dispersion, which ultimately prevents the network from continuing to train. A random initialization method is generally used to assign a weight value between –1 and 1 to solve this problem, but this method increases the uncontrollability of the network to a certain extent. In some cases, the performance is not good. On the basis of this finding, Glorot et al. proposed the Xavier initialization method [24] to make the variance of the output of each layer of the network as uniform as possible so that the gradient propagation does not cause gradient explosion or dispersion. Therefore, this study uses the Xavier initialization method to perform UNet weight initialization.

### 2.6. Vegetation Index Calculation

Vegetation indices are mainly used to reflect the difference between visible light, near-infrared reflection, and soil background. Vegetation indices can be used to quantitatively describe vegetation growth under certain conditions. Six kinds of visible vegetation indices were extracted from RGB images to study the influence of vegetation index on the results, and six multispectral vegetation indices were extracted from multispectral images. Table 1 lists the calculation formulas of various vegetation indices.

## 3. Results

### 3.1. Model Training

A total of 5 h 44 min and 5 h 36 min were spent on the training of the RGB image dataset and the multispectral (RGN) image dataset, respectively. During model training, the loss and dice coefficient were calculated on the training and validation datasets at the end of each epoch. Figure 5 and Figure 6 show the loss and dice coefficient curves of the RGB and multispectral (RGN) image datasets in the training and validation datasets, respectively. At the beginning of training, the loss and dice coefficient curves show a large oscillation phenomenon, which is more evident in multispectral image dataset training than that in the RGB image dataset training. This observation is related to the small mini-batch size set in this study. The network cannot find the convergence direction well at the early stage of training, but the network gradually becomes stable with continuous network training. In addition, the scale of network oscillations in the late training period is even smaller than that in the early training period because of the strategy of halving the learning rate every 10 epochs. As shown in Table 2, on the RGB image dataset, the dice coefficient of the validation dataset reaches the highest value, which is 0.9382, at the 75th epoch, the dice coefficient of the training dataset is 0.9468 at the same epoch. On the multispectral image dataset, the dice coefficient of the validation dataset reaches the highest value, which is 0.9222, at the 95th epoch, the dice coefficient of the training dataset is 0.9291 at the same epoch.

### 3.2. Model Prediction

The model with the highest accuracy in the validation dataset is selected as the final model and applied to the test dataset. Figure 7 illustrates the partial segmentation results in the test dataset, and Table 3 shows the average performance of the model on the test dataset. Intuitively, the model obtained by respectively applying the two datasets can basically segment the rice lodging part (white part of the segmentation result) on the image. The dice coefficient in Table 3 is above 0.92, which also confirms this point. However, from the difference between the two dataset results, the over-segmentation of the model obtained by training the multispectral image dataset is higher than that of the RGB image dataset. The loss value in Table 3 is well explained, that is, the result of the multispectral dataset is 0.1188, which is more than the result of the RGB data set.

Given that the resolution of the rice canopy image taken by the UAV is much larger than 320 × 320 pixels and the width and height are not multiples of 32, it cannot be directly input into the UNet model. The original large image should be block processed. In this study, the sliding window method is used to traverse the image. Each window has a size of 1280 × 1280 pixels. When the sliding window area exceeds the image area, the sliding window is filled with zeros. After the traversal is completed, the result of exceeding the area is discarded. In addition, given that the UNet model output is a probability description, that is, the pixel value of the output image is between 0 and 1, the network should be thresholded to convert it into a binary image. In this study, the threshold is set to 0.5. The pixel value of more than 0.5 is the lodged area, whereas the pixel value of less than 0.5 is the non-lodged area. Figure 8 and Figure 9 present the complete original large image, the corresponding segmentation result map and manually processed ground image. The results are compared with the artificially interpreted segmentation image to calculate the dice coefficient. The complete RGB and multispectral image segmentation results showing excellent dice coefficient results of 0.9626 and 0.9531, respectively.

The segmentation results indicate that the dice coefficients of two datasets are above 0.95 and effect is excellent. The uneven illumination due to the shadow of the trees on the left side of the image can be well segmented, explaining that the model can cope well with an uneven illumination problem.

### 3.3. Damage Assessment

Assessment of the damage areas by rice lodging is critical given the fact that it caused severe reductions in rice harvesting yield. In this section, we show how to apply the study to the damage assessment in practice. 

Since different farmers have different fields, it is first necessary to select the area of the field to be processed in the complete image. This step is done by manually selecting a polygon area, and the candidate area named mark mask (as shown in Figure 10) is obtained. Then the image corresponding to the mark mask is input into the model obtained in the study to obtain the prediction result image, and then the ratio of the lodging area is the ratio of the white area to the total area in the prediction result image. Agricultural damage assessment can be performed based on this ratio and can provide technical support for agricultural insurance claims. An example of a mark is presented in Figure 10, and the lodging area ratio is shown in Table 4.

### 3.4. Vegetation Indices as Inputs of UNet

This part of the study was divided into two cases. In the first case, the three vegetation indices are calculated and combined into a matrix of depth 3 as the input of UNet. The three visible vegetation indices are excess green (ExG), excess red (ExR), and visible-band difference vegetation index (VDVI), and the three multispectral vegetation indices are Normalized Difference Vegetation Index (NDVI), Ratio Vegetation Index (RVI), and Normalized Difference Water Index (NDWI). In the second case, all six vegetation indices are calculated and combined into a matrix of depth 6 as the input of UNet. Table 5 shows the average performance indicators of the prediction results of different datasets in both cases by retaining the other parameters for UNet training.

## 4. Discussion

### 4.1. Influence of Feature Selection on the Recognition Accuracy of Lodged Rice

As shown in Table 5, the results of Case 1 (three kinds of indices) are slightly better than that of Case 2 (six kinds of indices). However, in the comparative analysis of Table 3 and Table 5, whether in Cases 1 or 2, the results obtained using the vegetation indices as UNet input are slightly worse than the results obtained by directly using the original image as UNet input. Generally speaking, the vegetation index is obtained by algebraic operation between different bands, which is basically a type of a feature description of the image. In the study, UNet was used to extract features of UAV images, and these features after band calculation were considered as UNet input is redundant. In addition, if the feature obtained by the algebraic operation between different bands does not fully characterize the original image, some special information is lost to some extent, which may eventually cause the degradation of network performance. Moreover, the result of Case 1 is slightly better than that of Case 2 under certain conditions, that is the dice coefficient (0.9396) of Case 1 is better than the dice coefficient (0.9348) of Case 2 in RGB dataset. It’s important to note that the higher the amount of input data there is, the better the achieved effect is. The excess data may interfere with other data in distribution and characterization of rice lodging. This phenomenon eventually leads to pessimistic results.

### 4.2. Influence of Different Sensors on the Recognition Accuracy of Rice Lodging

The images of rice canopy were obtained using a high-resolution digital camera and a multispectral camera with near-infrared bands. Then UNet models were established to predict the lodging of actual paddy fields through canopy images. Figure 11 shows two performance comparisons of the image dataset acquired by the sensor under different conditions (original image, three vegetation indices, and six vegetation indices) in solving the aforementioned problems. The RGB dataset loss results are smaller than that of the multispectral dataset, and the dice results are larger than those of the multispectral dataset, that is, the performance of the model trained on the RGB dataset is better than that of the multispectral dataset, indicating that a high-resolution digital camera is suitable for the processing of rice lodging image segmentation tasks.

Currently, the point cloud technology also is introduced into crops lodging detection. Wilke et al. [37] built a height model based on a point cloud to extract lodging information. Compared with RGB and multispectral sensors, the analysis of the lodging information based on the point cloud has the advantage of high precision, but the process of establishing the point cloud is complicated and the calculation amount is large [7,38,39,40]. However, it is fast and convenient to directly extract rice lodging information by using image information. In the example of this study, the accuracy can meet the application demand. Therefore, the method of extracting lodging information by using image information is extremely suitable for a scene with high speed requirements. Therefore, in the actual application environment, such as the agricultural insurance claims process, it is necessary to evaluate the crop lodging rate. Compared with the point cloud-based method, the method proposed in this study has great advantages in price and operability.

### 4.3. Influence of Different External Environmental Factors on the Recognition Accuracy of Rice Lodging

Compared to other machine learning algorithms, the end-to-end feature of the UNet architecture allows us to focus on the input and output of the task without having to extract complex features from the input data, which gives us the ability to iterate quickly while processing the task. In this study, we only need to pay attention to the input of image data and the output of the rice lodging assessment results, even if there are some differences between the input image data (such as different light, different nitrogen application, different rice varieties and other external environmental factors affect the image data), we only need to input image data with different differences into Unet for model updating. Which makes the method proposed in this study suitable for the real environment and is highly robust to different external factors.

## 5. Conclusions

In this study, a DJI Phantom4 UAV platform equipped with a high-resolution digital camera and a multispectral camera is used to obtain the RGB and multispectral images of rice lodging canopy. The images are preprocessed by splicing, cropping, and augmenting to generate an image dataset of 5000 images, and the images are then used to train the UNet model. The results show that the dice coefficients on the RGB and multispectral datasets reach 0.9442 and 0.9284, respectively. The model is applied to the complete image acquired by the UAV, and the output result is thresholded to obtain the segmented image. The results show that the dice coefficients on the RGB and multispectral images reach 0.96262 and 0.9531, respectively. The model is applied to assess the damage, and the rice lodging ratios are calculated to be 54.71% and 61.26% in the RGB and multispectral images on the example field, respectively. In addition, this study discusses the calculation of the vegetation index and the effect of different sensors on the UNet results. This study obtains optimal results by using a high-resolution digital camera to obtain images and directly input their original images into UNet.

This study proposes a new workflow pipeline for rice lodging assessment in high-throughput plant phenotyping scenarios. It can provide important methodological reference for large-area, high-efficiency and low-cost rice lodging monitoring research, and provide decision support for agricultural insurance and other fields.

## Figures and Tables

**Figure 1 sensors-19-03859-f001:**
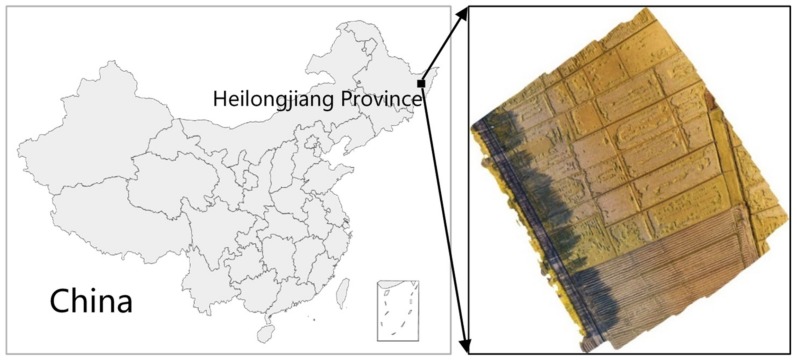
Schematic diagram of the study area.

**Figure 2 sensors-19-03859-f002:**
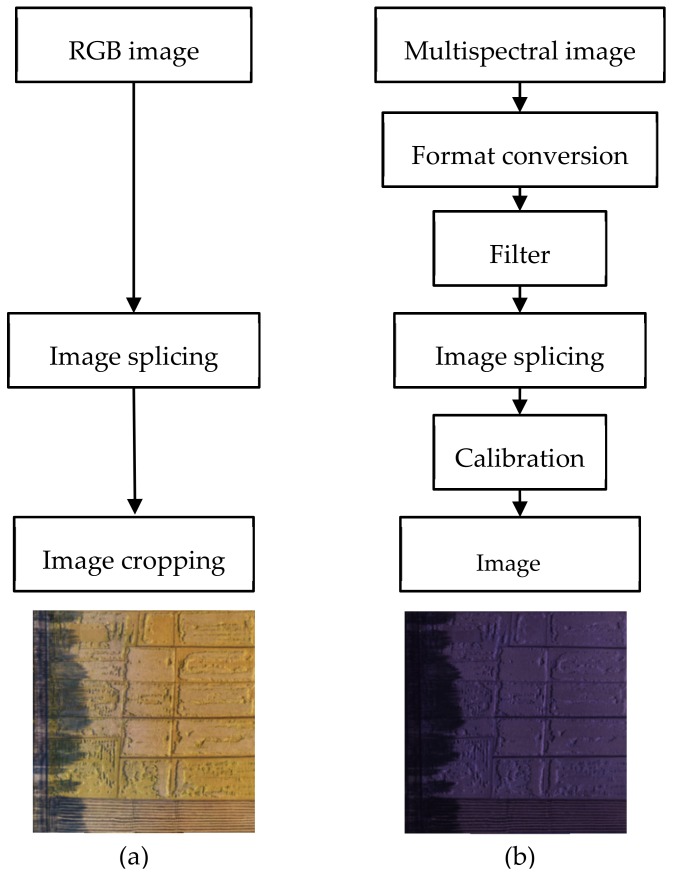
Data preprocessing: (**a**) the resulting RGB image; (**b**) the resulting multispectral image.

**Figure 3 sensors-19-03859-f003:**
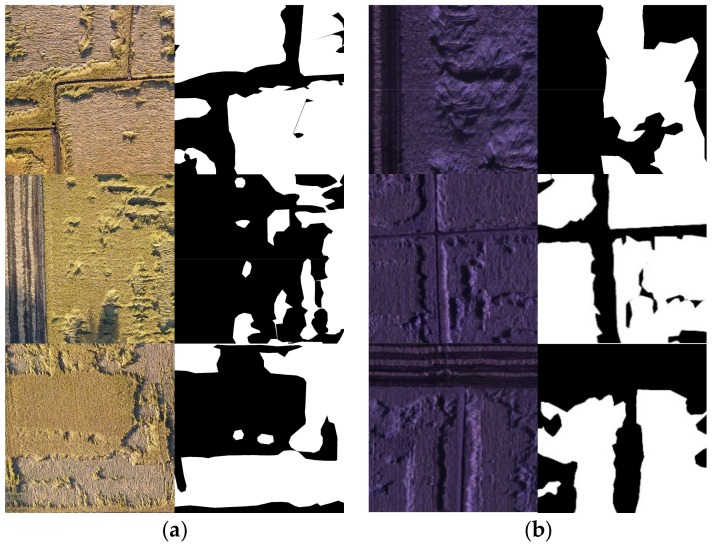
Sample dataset, the manually processed ground images are displayed as binary image: (**a**) RGB image dataset, displayed as color image; (**b**) multispectral image dataset, displayed as color image and pseudo color image.

**Figure 4 sensors-19-03859-f004:**
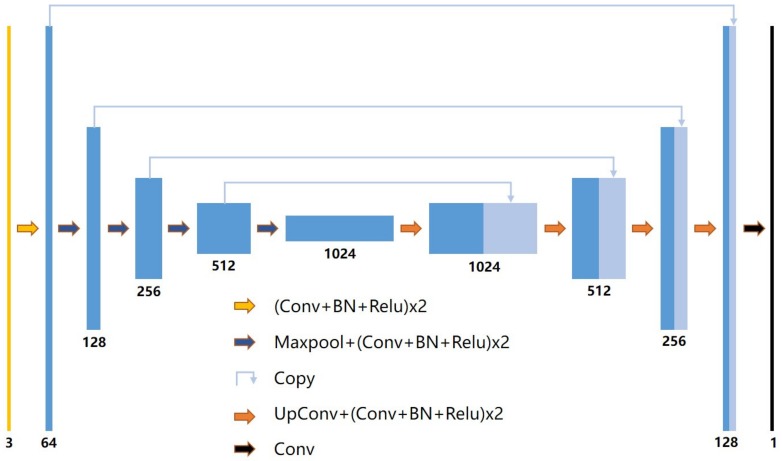
Improved UNet architecture. Conv is a convolutional layer (convolution kernel is 3 × 3 and padding is 1), BN is batch normalization, Relu is the activation layer, Maxpool is the maximum pooling layer, UpConv is the transposed convolution.

**Figure 5 sensors-19-03859-f005:**
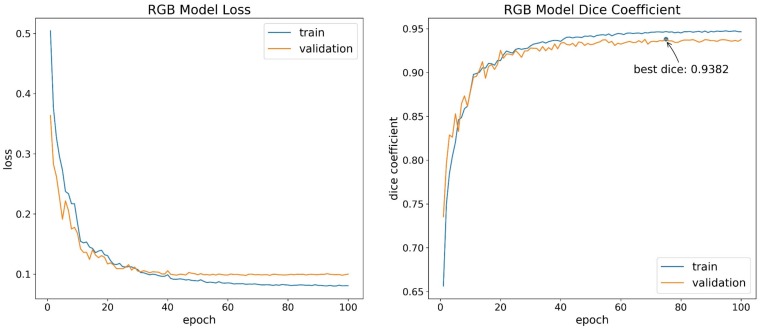
Loss and dice curves of the RGB image dataset model.

**Figure 6 sensors-19-03859-f006:**
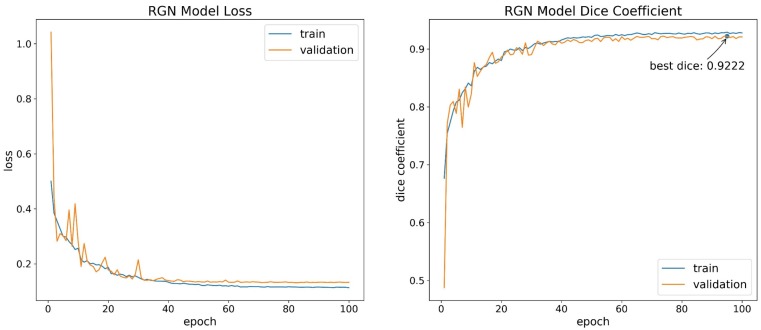
Loss and dice curves of the Red, Green, NIR (RGN) image dataset model.

**Figure 7 sensors-19-03859-f007:**
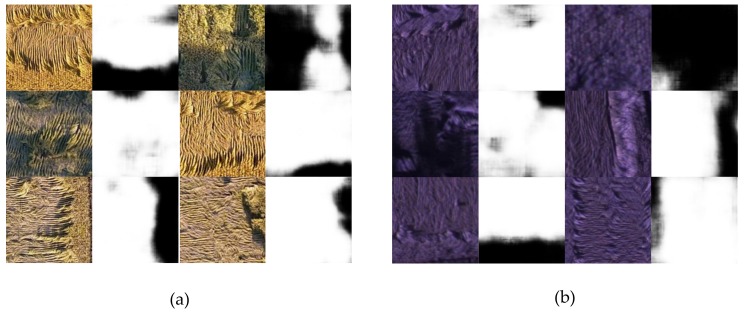
Test dataset segmentation results: (**a**) RGB image test dataset segmentation results; (**b**) multispectral image test dataset segmentation results.

**Figure 8 sensors-19-03859-f008:**
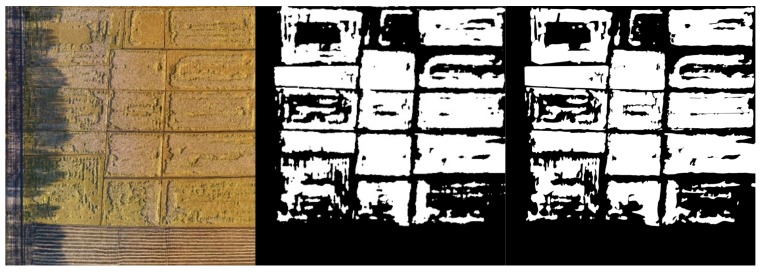
RGB original image (**left**), segmentation result image (**middle**) and manually processed ground image (**right**).

**Figure 9 sensors-19-03859-f009:**
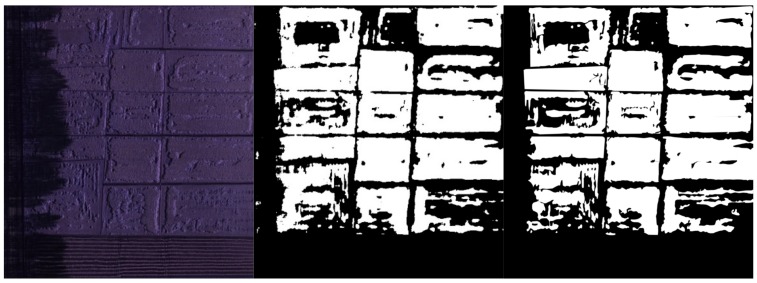
Multispectral original image (**left**), segmentation result image (**middle**) and manually processed ground image (**right**).

**Figure 10 sensors-19-03859-f010:**
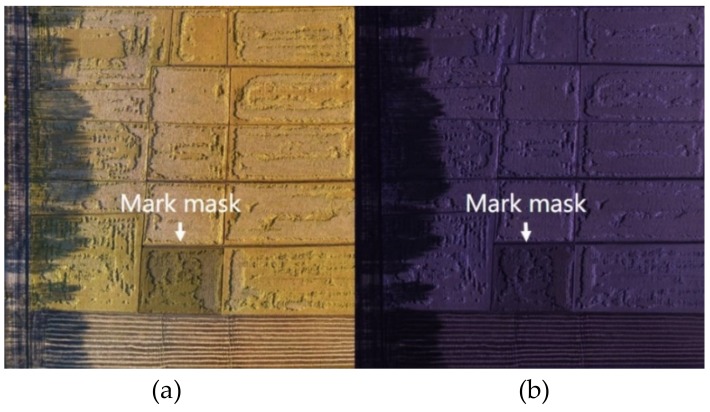
Field marker: (**a**) field marker of RGB image; (**b**) field marker of multispectral image.

**Figure 11 sensors-19-03859-f011:**
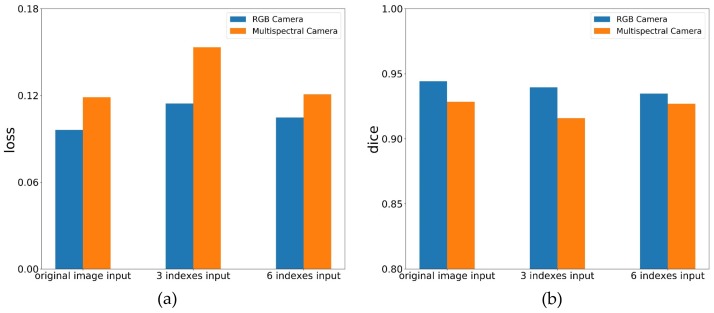
Comparison of different sensor applications: (**a**) loss comparison of different sensor applications; (**b**) dice comparison of different sensor applications.

**Table 1 sensors-19-03859-t001:** Vegetation Index table.

Type	Vegetation Index	Calculation Formula	References
Visible Vegetation index	ExG (Excess Green)	*2g-r-b*	[25]
	ExR (Excess Red)	*1.4R-G*	[26]
	VDVI (Visible-Band Difference Vegetation Index)	*(2G-R-B)/(2G+R+B)*	[27]
	RGRI (Red-Green Ratio Index)	*R/G*	[28]
	NGRDI (Normalized Green-Red Difference Index)	*(G-R)/(G+R)*	[29]
	ExGR (Excess Green minus Excess Red)	*ExG-ExR*	[30]
Multispectral Vegetation index	NDVI (Normalized Difference Vegetation Index)	*(NIR-R)/(NIR+R)*	[31]
	RVI (Ratio Vegetation Index)	*NIR/R*	[32]
	NDWI (Normalized Difference Water Index)	*(G-NIR)/(G+NIR)*	[33]
	DVI (Difference Vegetation Index)	*NIR-R*	[34]
	PVI (Perpendicular Vegetation Index)	$\frac{NIR-10.489R-6.604}{\sqrt{1+10.489^{2}}}$	[35]
	SAVI(Soil-Adjusted Vegetation Index)	$\frac{\left(1+0.5\right)\left(NIR-R\right)}{NIR+R+0.5}$	[36]

Note: *R*, *G*, *B*, and *NIR* are the pixel values of the image red, green, blue and near-infrared, respectively, and *r*, *g*, and *b* are the normalized pixel values of the image red, green, and blue channels, respectively.

**Table 2 sensors-19-03859-t002:** Dice coefficient of the training dataset and validation dataset.

Dataset	Training Dice Coefficient	Validation Dice Coefficient
RGB	0.9468	0.9382
Multispectral	0.9291	0.9222

**Table 3 sensors-19-03859-t003:** Model average performance.

Dataset	Loss	Dice Coefficient
RGB	0.0961	0.9442
Multispectral	0.1188	0.9284

**Table 4 sensors-19-03859-t004:** Ratio of lodging.

Dataset	Lodging Ratio
RGB	54.71%
Multispectral	61.26%

**Table 5 sensors-19-03859-t005:** Model average performance.

Vegetation Index Type	Dataset	Loss	Dice Coefficient
3 kinds of indices	RGB	0.1144	0.9396
	Multispectral	0.1534	0.9158
6 kinds of indices	RGB	0.1047	0.9348
	Multispectral	0.1207	0.9270

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
