# Peer review of "Use of Unmanned Aerial Vehicle Imagery and Deep Learning UNet to Extract Rice Lodging"

_sensors, 2019, doi:10.3390/s19183859_

Round 1

Reviewer 1 Report

Dear Author,

I have read your paper with great interest because I have seen other similar studies.

My comments are following:

Using UNet algorithm is good but I would suggest you should also do point clouds based analysis. I mean you should check what is the difference in the point clouds of lodged rice field and normal rice field. This will add novelty to your research. Based on the outcomes you can certainly say which method will be more appropriate.

You should write more about the accuracy of the MAPIR camera.

Did you cover the whole study area with 1 flight or multiple flights?

Why did you choose 100m flight height?

Other Comments:

16: measurements which is found to be time-consuming labor-intensive, and cost-intensive.

26: The findings of this study are useful for investigations on rice lodging assessment using

32: Rice is one of the world’s three major food crops besides from ___ and___,

79: In this study, DJI Phantom 4 UAV Platform equipped with high resolution digital and multispectral camera was used.

91: a 12-bit RAW format and captured in two-second interval.

113: The UNet

122: In this study, the network was improved to make it suitable for the segmentation of rice lodging

206 - 209: The complete RGB and multispectral image segmentation results showing excellent dice coefficient results of 0.9626 and 0.9531, respectively. 

217: Assessment of the damage areas by rice lodging is critical given the fact that it causing severe reductions in rice harvesting yield. 

218: Artificial segmentation was

229: Vegetation indices are mainly

230: Vegetation indices can be used to

260: The images of rice canopy were obtained using a high-resolution digital camera and a multispectral camera with near-infrared bands. Then UNet models were established to predict the lodging of actual paddy fields through canopy images.   

272: In this study, a DJI Phantom4 UAV platform equipped with a high-resolution digital camera 272 and a multispectral camera is used to obtain the RGB and multispectral images of rice lodging canopy.

*“In this study” has been used many times and consecutively in the last two paragraphs.

Author Response

Please see the attachment, we have made targeted changes to the questions raised by the reviewer. However, due to the difference in understanding, the manuscript may not be perfected. Please provide us with the opportunity to continue to revise it. We will seriously supplement and improve it. Thanks again to the reviewer and editor for their valuable suggestions.

Reviewer 2 Report

Comments

1Why was the experimental site chosen in Qixing Farm? Does the authors have no experimental base in Anhui province?

2Why chose a flight altitude of 100 meters? What was the relationship between different flight heights and resolution?

3The resolutions of both digital camera and multispectral camera were 4000×3000 pixels, Why were the image resolutions obtained in the image processing section 7337×7660 and 6754×7370 pixels, respectively?

4The coefficients of RGB and RGN training set were 0.9382 and 0.9222 respectively, while the coefficients of prediction were 0.9442 and 0.9284 respectively. Will the prediction accuracy be higher than the training accuracy?

5What were the Ground truth images in figures 8 and 9? Why were the two pictures different

6How was the rice lodging area extracted? How was the ratio of lodging area calculated?

7How to distinguish the lodging difference of different rice varieties? How to distinguish the effect of different nitrogen application on lodging? These are very important in practical applications.

8How to determine the boundary between rice lodging area and non-lodging area when extracting rice lodging area?

Author Response

(The authors gave the same response as above.)

Reviewer 3 Report

Dear Editor, I had the opportunity to review the manuscript "Use of Unmanned Aerial Vehicle Imagery and Deep Learning UNet to Extract Rice Lodging". The article deals with a recent topic and within the scope of the Journal Sensors. However, I have some important concerns before it can be considered for publication.

Abstract:

- define acronyms the first time it is mentioned in the text;

Keywords:

- add the scientific name of the rice;

Introduction:

- put the scientific name on line 32;

- the first paragraph needs several citations to support the claims made by the authors;

- It is necessary to include in the penultimate paragraph of this item recent research results and why this research is different from the others;

 Materials and Methods:

- I suggest creating a topic "Statistical analysis";

- It is necessary to add some more statistical procedure to validate the obtained results. Only the coefficient coefficient is not enough; I suggest the coefficient of determination (R²). Without the addition of any statistical technique the manuscript cannot be accepted for publication;

- Authors need to separate the dataset in training (80% of data) and validation (20% of data). This allows you to evaluate the different deep learning architectures that can be tested;

- It is necessary to provide in which software the analyzes were performed. Is it free software? Can authors make the script available?

Results

- The results may be expanded according to the changes made in the statistics.

Discussion

- Table 3 is in inappropriate place. Move it to M&M;

- Table 4 and Figure 5 should be moved to R&D;

Conclusions

- This item should be written more succinctly and respond to the proposed objectives.

References

- I believe 27 references are too little for a scientific manuscript. Authors should support their search for papers in the field.

Author Response

(The authors gave the same response as above.)

Reviewer 4 Report

Use of Unmanned Aerial Vehicle Imagery and Deep Learning UNet to Extract Rice Lodging

The manuscript reports an investigation on the use of UNnet models to predict the lodging of rice using RGB and Multispectral camera. Despite the promising results, the paper needs to be improved and better organised. Although major grammatical mistakes are missing, the manuscript would benefit from English review.

The methodology applied is not entirely defined: it does not mention, indeed, the use of vegetation indices, which appears only in the Discussion section. A significant part of the methodology is explained in the Results and Discussion sections.

The results of the model used are clearly explained, but the results of the analysis of the vegetation indices are mentioned only in the Discussion section.

The interpretation of the results is limited to lines 260-268, but still, these lines are a mix of methodology, results and discussion.

The conclusions report a summary of the results without providing any indication for future development.

Ethics: the manuscript complies with standard COPE ethical guidelines.

Line 54: there is the need of a reference for Li Zongnan. The reference is in the next sentence and the connection is not clear.

Line 58: the word “mature” doesn’t seem appropriate

Line 73: the unity of measurement “mu” should be replaced by an international one

Line 84: the authors refer to a single image, but from the text of the manuscript it looks clear that different cameras were used

Line 85: does “level 2” refer to Beaufort scale? Please specify

Line 127: the readers would benefit from some explanation of the technical abbreviation (i.e. BN, relu,…) at this point. The abbreviations are explained only in lines 134-136

Lines 146-168: these lines should be moved to the Methods section

Line 229 and line 231: “Vegetation index” should be replaced by “Vegetation indices”

Line 231: “kinds” should be replaced by “kind”

Lines 228-258: this section discusses the results of the vegetation indices, but the Methods section and the Results section don’t mention them.

Lines 255-256: this sentence is not clear for me

The reference of the figures 1,2, 3 is missing in the manuscript

The captation of Figure 4 needs some improvement (abbreviations should be explained)

Author Response

(The authors gave the same response as above.)

Round 2

Reviewer 1 Report

Dear Author 

Thank you very much for revision and sharing the revised version of the manuscript. I agree with your reply but still if you have time and resources try to compare the point-cloud based analysis and simple RGB based analysis to get the information about rice lodging. It would enhance the scinetific knowledge of your manuscript. 

Author Response

We will further analyze the impact of more different factors on the results in future research, including analysis of point-cloud data. Thank you very much for reviewing of the manuscript.

Reviewer 2 Report

1、The authors have yet to clarify the relationship between flight altitude and image resolution.

2、Theoretically,the prediction accuracy should not be higher than the modeling accuracy(Point 4).

3、What is the ground truth image? If it's manually labelled, it shouldn't be called a ground truth image, it should be called a manually processed ground image(Point 5)

4、If the lodging area is calculated by manually selected, the method in the paper is meaningless(Point 6)

5、As for the effect of different nitrogen application on the image, the authors did not make clear, because the color and texture features of the image may be very similar between the lodging area of one nitrogen application rate and the non-lodging area of another nitrogen application rate. This problem also exists between different rice varieties (Point 7)

Author Response

(The authors gave the same response as above.)

Reviewer 3 Report

The authors made the corrections and this manuscript can be accept in present form

Author Response

Thank you very much for reviewing of the manuscript.

Reviewer 4 Report

Use of Unmanned Aerial Vehicle Imagery and Deep Learning UNet to Extract Rice Lodging

The structure of the paper has been improved. The methodology is now sufficiently explained and the results are reported in a logical sequence.

The discussion is still poor but some comments to the results are provided in the "Results" section.

Despite it's simple structure, the paper can add some suggestions for scientific research.

Check for spacing in Table 1.

Author Response

(The authors gave the same response as above.)
